# Efficient Certified Defenses Against Patch Attacks on Image Classifiers

**Jan Hendrik Metzen & Maksym Yatsura**
Bosch Center for Artificial Intelligence, Robert Bosch GmbH
Robert-Bosch-Campus 1, 71272 Renningen, Germany
{janhendrik.metzen,maksym.yatsura}@de.bosch.com

## Abstract

Adversarial patches pose a realistic threat model for physical world attacks on autonomous systems via their perception component. Autonomous systems in safety-critical domains such as automated driving should thus contain a fail-safe fallback component that combines certifiable robustness against patches with efficient inference while maintaining high performance on clean inputs. We propose BAGCERT, a novel combination of model architecture and certification procedure that allows efficient certification. We derive a loss that enables end-to-end optimization of certified robustness against patches of different sizes and locations. On CIFAR10, BAGCERT certifies 10.000 examples in 43 seconds on a single GPU and obtains 86% clean and 60% certified accuracy against $5 \times 5$ patches.

## 1 Introduction

Adversarial patches (Brown et al., 2017) are one of the most relevant threat models for attacks on autonomous systems such as highly automated cars or robots. In this threat model, an attacker can freely control a small subregion of the input (the "patch") but needs to leave the rest of the input unchanged. This threat model is relevant because it corresponds to a physically realizable attack (Lee & Kolter, 2019): an attacker can print the adversarial patch pattern, place it in the physical world, and it will become part of the input of any system whose field of view overlaps with the physical patch. Moreover, once an attacker has generated a successful patch pattern, this pattern can be easily shared, will be effective against all systems using the same perception component, and an attack can be conducted without requiring access to the individual system. This makes for instance attacking an entire fleet of cars of the same vendor feasible.

While several empirical defenses were proposed (Hayes, 2018; Naseer et al., 2019; Selvaraju at al., 2019; Wu et al., 2020)), these only offer robustness against known attacks but not necessarily against more effective attacks that may be developed in the future (Chiang et al., 2020). In contrast, certified defenses for the patch threat model (Chiang et al., 2020; Levine & Feizi, 2020; Zhang et al., 2020; Xiang et al., 2020) allow guaranteed robustness against all possible attacks for the given threat model. Ideally, a certified defense should combine high certified robustness with efficient inference while maintaining strong performance on clean inputs. Moreover, the training objective should be based on the certification problem to avoid post-hoc calibration of the model for certification.

Existing defenses do not satisfy all of these conditions: Chiang et al. (2020) proposed an approach that extends interval-bound propagation (Gowal et al., 2019) to the patch threat model. In this approach, there is a clear connection between training objective and certification problem. However, certified accuracy is relatively low and clean performance severely affected (below $50\%$ on CIFAR10). Moreover, inference requires separate forward passes for all possible patch positions and is thus computationally very expensive. Derandomized smoothing (Levine & Feizi, 2020) achieves much higher certified and clean performance on CIFAR10 and even scales to ImageNet. However, inference is computationally expensive since it is based on separately propagating many differently ablated versions of a single input. Moreover, training and certification are disconnected and a separate tuning of parameters of the post-hoc certification procedure on some hold-out data is required, a drawback shared also by Clipped BagNet Zhang et al. (2020) and PatchGuard (Xiang et al., 2020).

In this work, we propose BAGCERT, which combines high certified accuracy ($60\%$ on CIFAR10 for $5 \times 5$ patches) and clean performance ($86\%$ on CIFAR10), efficient inference (43 seconds on a single GPU for the 10.000 CIFAR10 test samples), and end-to-end training for robustness against patches of varying size, aspect ratio, and location. BAGCERT is based on the following contributions:

- We propose three different conditions that can be checked for certifying robustness. One of these corresponds to the condition proposed by Levine & Feizi (2020). However, we show that an alternative condition improves certified accuracy of the same model typically by roughly 3 percent points while remaining broadly applicable.
- We derive a loss function that directly optimizes for certified accuracy against a uniform distribution of patch sizes at arbitrary positions. This loss corresponds to a specific type of the well known class of margin losses.
- Similarly to Levine & Feizi (2020), we classify images via a majority voting over a large number of predictions that are based on small local regions of a single input. However, the proposed model achieves this via a single forward-pass on the unmodified input, by utilizing a neural network architecture with very small receptive fields, similar to BagNets (Brendel & Bethge, 2019). This enables efficient inference with surprisingly high clean accuracy and was concurrently proposed by Zhang et al. (2020) and Xiang et al. (2020).

## 2 RELATED WORK

**Adversarial Patch Attacks** Vulnerability of image classifiers to adversarial patch attacks was first demonstrated by Brown et al. (2017). They show that a specifically crafted physical adversarial patch is able to fool multiple ImageNet models into predicting the wrong class with high confidence. Numerous patch attacks were proposed for object detection (Liu et al., 2019; Lee & Kolter, 2019; Thys et al., 2019; Huang et al., 2019) and optical flow estimation (Ranjan et al., 2019). The versatility of these attacks allows them to perform efficiently in the black box setup (Croce et al., 2020) as well as suppressing detected objects in a scene without overlapping any of them (Lee & Kolter, 2019). Following Athalye et al. (2018b), adversarial patches can be printed out and placed in the physical world to fool different models independently from the scaling, rotation, brightness and other visual transformations. These factors make adversarial patch attacks a non-negligible threat for the safety-critical perception systems (Thys et al., 2019).

**Heuristic Defenses Against Patch Attacks** Several heuristic defenses against adversarial patches such as digital watermarking (Hayes, 2018) or local gradient smoothing (Naseer et al., 2019) have been proposed. However, similarly to the results obtained for the norm-bounded adversarial attacks (Athalye et al., 2018a), it was demonstrated that these defenses can be easily broken by white-box attacks which account for the pre-processing steps in the optimization procedure (Chiang et al., 2020). The role of spatial context in the object detection algorithms which makes them vulnerable to the patch attacks was investigated by Saha et al. (2019) and an empirical defense based on Grad-CAM (Selvaraju et al., 2019) was proposed. Existing augmentation techniques based on adding Gaussian noise patch (Lopes et al., 2019) or a patch from a different image (Yun et al., 2019) increase robustness against occlusions caused by adversarial patches. Wu et al. (2020) propose a defense that uses adversarial training to increase robustness against occlusion attacks.

**Certified Defenses** Evaluating defense methods using their performance against empirical attacks can lead to the false sense of security since stronger adversaries might be developed in the future that break the defenses (Athalye et al., 2018a; Uesato et al., 2018). Therefore, it is important to have guarantees of robustness. Numerous works were proposed in the field of certified robustness ranging from complete verifiers finding the worst-case adversarial examples exactly (Huang et al., 2017; Tjeng & Tedrake, 2017) to faster but less accurate incomplete methods that provide an upper bound on the robust error (Gehr et al., 2018; Wong & Kolter, 2018; Wong et al., 2018; Gowal et al., 2019). Another line of work is based on Randomized Smoothing (Lecuyer et al., 2019; Li et al., 2019; Cohen et al., 2019), which exhibits strong empirical results and scales to ImageNet, however at the cost of increasing inference time by orders of magnitude. Certified defenses crafted for the patch attacks were first proposed by Chiang et al. (2020). They adapt the IBP method (Gowal et al., 2019) to the patch threat model. Although their approach allows to obtain robustness guarantees, it only scales to small patches and causes a significant drop in clean accuracy. Levine & Feizi (2020)

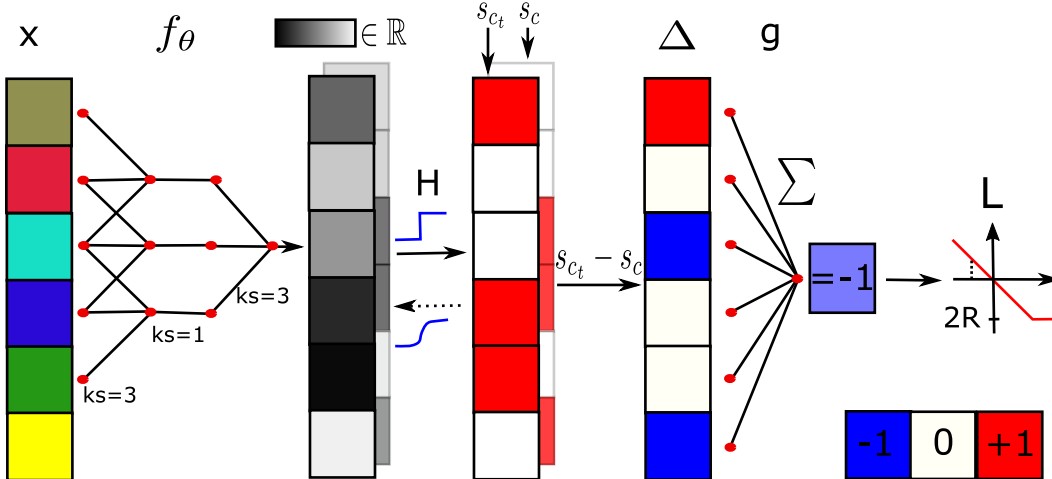

Figure 1: Illustration of BAGCERT training for a 1D input and two classes. An input $X$ is processed by region scorer $f_\theta$, consisting of a 3-layer CNN with kernel sizes 3, 1, and 3. The resulting continuous region scores are passed through a Heaviside step function (replaced by a sigmoid in the backward pass) to obtain binary region scores $s$ for every class. The differences $\Delta$ between true and non-true class scores are then processed by spatial aggregation $g$, in this case simply summing them via $g = g_\Sigma$. The resulting value is maximized by passing it into margin loss $L$.

proposed (de)randomized smoothing: they train a base classifier for classifying images where all but a small local region is ablated. At inference time, many (or even all) possible ablations are classified and a majority vote determines the final classification. If this majority vote is with sufficient margin, the decision is provable robust against patch attacks because a patch will be completely ablated in most of the inputs and can thus only influence a minority of the votes. This method provides significant accuracy improvement when compared to (Chiang et al., 2020) and allows training and certifying ImageNet models. However, its inference in block-smoothing mode is computationally expensive. A last line of work is based on using models with small receptive fields such as BagNets (Brendel & Bethge, 2019): Zhang et al. (2020) apply a clipping function while Xiang et al. (2020) apply a "detect-and-mask" filter to the logits of pretrained BagNets before global averaging. Using small receptive fields limits the number of region scores affected by a local patches while clipping and masking ensure that few very large region scores cannot dominate the global average. We note that these approaches do not train models directly for certified robustness but rather achieve it by applying post-hoc procedures that come with additional hyperparameters that require careful tuning.

## 3 METHOD

We introduce BAGCERT, a framework which consists of novel conditions for certifying robustness, a specific model architecture, and a new end-to-end training procedure. BAGCERT allows end-to-end training of classifiers whose robustness against adversarial patch attacks can be certified efficiently. We outline our approach for the task of image classification but note that it can be extended to other tasks with grid-structured inputs. We refer to Figure 1 for an illustration of the training phase and to Figure 5 in the supplementary material for an illustration of certification of BAGCERT.

**Threat Model**  We consider a threat model in which an attacker can conduct an image-dependent patch attack. Let $x \in [0,1]^{w_{in} \times h_{in} \times c_{in}}$ be an input image of resolution $w_{in} \times h_{in}$ with $c_{in}$ channels. Let $p$ be a patch and $l$ be a region of an image $x$ having the same size as patch $p$. We denote a set of feasible regions $l$ as $\mathcal{L}$. For example, for a patch $p \in [0,1]^{n \times c_{in}}$ consisting of $n$ pixels, $\mathcal{L}$ could be the set of all $w_p \times h_p$ rectangular regions $l$ of an image $x$ with $w_p \cdot h_p = n$. We define an operator $A$ such that $A(x, p, l)$ is the result of placing a patch $p$ onto an image $x$ over a region $l$. We assume that the attacker has white-box knowledge of the model and conducts an input-dependent attack, that is attack region $l$ and inserted patch $p$ can be chosen for every input independently.

## 3.1 CERTIFICATION

We base our method for certification on assuming a certain structure of the classifier. More specifically, we decompose the classifier into two components:

- A region scorer $f_\theta$ that maps from inputs $x$ to region scores $s \in \{0, 1\}^{w_{out} \times h_{out} \times c_{out}}$, where $w_{out} \times h_{out}$ is the output resolution, $c_{out}$ is the number of classes, and $\theta$ are trainable parameters. Please note that we allow $\sum_c s_{i,j,c} \neq 1$.

- A spatial aggregator $g$ that maps from region scores $s$ to (global) class scores $S \in [0, 1]^{c_{out}}$. In this work, we restrict $g$ to be monotonically increasing, that is: for class $c$ and two patch score maps $s^{(1)}$ and $s^{(2)}$ with $s_{i,j,c}^{(1)} \geq s_{i,j,c}^{(2)} \, \forall i, j$, we require $g(s^{(1)})_c \geq g(s^{(2)})_c \, \forall c$.

Generally, we base certification on upper bounding the effect of an actual attack in the threat model. For this, we only exploit architectural properties of $f$ and $g$ that are valid for any choice of model parameters $\theta$. More specifically, we only exploit the output dependency map $R$ of $f$, which we define as $R(l) = \{(i, j) \mid \exists x, \theta, p : f_\theta(A(x, p, l))_{i,j} \neq f_\theta(x)_{i,j}\}$. Informally, $R(l)$ is the set of all indices of the score map that can be affected by a patch applied at region $l$, for any choice of input $x$, patch $p$, and parameters $\theta$. That is: the set of all outputs of $f_\theta$ whose receptive fields overlap with $l$. We discuss options for $f$ and the resulting $R$ in Section 3.2.

For input $x$ with class label $c_t$ and $s = f_\theta(x)$, we define the "worst-case" score map $s^{wc}(s, l)$ as

$$
s_{i,j,c}^{wc}(s, l) = \begin{cases} s_{i,j,c} & \text{if } (i, j) \notin R(l) \\ 1 & \text{if } (i, j) \in R(l) \wedge c \neq c_t \\ 0 & \text{if } (i, j) \in R(l) \wedge c = c_t \end{cases}
$$

Moreover, we define $\Delta_{i,j,c} = s_{i,j,c_t} - s_{i,j,c}$ and similarly $\Delta_{i,j,c}^{wc} = s_{i,j,c_t}^{wc} - s_{i,j,c}^{wc}$. It follows directly that $\Delta_{i,j,c}^{wc} = \Delta_{i,j,c} \, \forall (i, j) \notin R(l)$ and $\Delta_{i,j,c}^{wc} = -1 \, \forall (i, j) \in R(l), c \neq c_t$.

For certifying robustness in the threat model for input $x$ with class label $c_t$, we need to show $g(f_\theta(A(x, p, l)))_{c_t} > g(f_\theta(A(x, p, l)))_c \, \forall c \neq c_t \, \forall l \in \mathcal{L} \, \forall p$. For this, it suffices to check

**Condition 3.1.** $g(s^{wc}(s, l))_{c_t} > g(s^{wc}(s, l))_c \, \forall c \neq c_t, \forall l \in \mathcal{L}$

*Proof.* Consider arbitrary $l \in \mathcal{L}$ and $p$ and let $s^{adv} = f_\theta(A(x, p, l))$. With $s_{i,j,c}^{adv} \in \{0, 1\}$ we obtain[1]

$$
s_{i,j,c}^{adv} \begin{cases} = s_{i,j,c}^{wc}(s, l) = s_{i,j,c}(s, l) & \text{if } (i, j) \notin R(l) \\ \leq s_{i,j,c}^{wc}(s, l) = 1 & \text{if } (i, j) \in R(l) \wedge c \neq c_t \\ \geq s_{i,j,c}^{wc}(s, l) = 0 & \text{if } (i, j) \in R(l) \wedge c = c_t \end{cases}
$$

With $g$ being monotonically increasing, we obtain $g(s^{adv})_{c_t} \geq g(s^{wc}(l))_{c_t}$ and for all $c \neq c_t$ $g(s^{wc}(l))_c \geq g(s^{adv})_c$. Condition 3.1 implies $g(s^{adv})_{c_t} > g(s^{adv})_c \, \forall c \neq c_t$. □

Checking the Condition 3.1 requires one forward-pass through $f_\theta$ to obtain $s = f_\theta(x)$ and $|\mathcal{L}|$ times the construction $s^{wc}(s, l)$ and the evaluation of $g$. We now consider a special case where this can be implemented very efficiently.

### 3.1.1 SPATIAL SUM AGGREGATION

For the case $g = g_\sum(s) = \sum_{i=1,j=1}^{w_{out}, h_{out}} s_{i,j}$, Condition 3.1 simplifies to

**Condition 3.2.** $\min_{c \neq c_t} \sum_{i,j \notin R(l)} \Delta_{i,j,c} > |R(l)| \quad \forall l \in \mathcal{L}$

---

[1]We would like to note that these are "trivial" lower and upper bounds for $s^{adv}$ and we see the potential to improve upon these bounds in future work, for instance by relaxing $s \in [0, 1]^{w_{out} \times h_{out} \times c_{out}}$ and applying interval bound propagation (Gowal et al., 2019). However, the proposed simple bounds have the advantage of not requiring additional forward passes through the model and thus being computationally efficient.

*Proof.* For all $c \neq c_t$, we exploit $\forall (i,j) \in R(l) : \Delta_{i,j,c}^{wc} = -1$. With Condition 3.2, we obtain

$$g_{\sum}(s^{wc}(l))_{c_t} - g_{\sum}(s^{wc}(l))_c = \sum_{i=1,j=1}^{w_{out},h_{out}} s_{i,j,c_t}^{wc} - \sum_{i=1,j=1}^{w_{out},h_{out}} s_{i,j,c}^{wc} = \sum_{i=1,j=1}^{w_{out},h_{out}} \Delta_{i,j,c}^{wc}$$

$$= \sum_{i,j \notin R(l)} \Delta_{i,j,c}^{wc} + \sum_{i,j \in R(l)} \Delta_{i,j,c}^{wc}$$

$$= \sum_{i,j \notin R(l)} \Delta_{i,j,c} - |R(l)| > 0. \qquad \square$$

We note that $\sum_{i,j \notin R(l)} \Delta_{i,j,c} = \sum_{i=1,j=1}^{w_{out},h_{out}} \Delta_{i,j,c} - \sum_{i,j \in R(l)} \Delta_{i,j,c_t}$. For the special case that all $R(l)$ are rectangular, $\sum_{i,j \in R(l)} \Delta_{i,j,c}$ can be computed efficiently for all $l \in \mathcal{L}$ simultaneously via integral images/summed-area tables (Crow, 1984). For instance, $R(l)$ is rectangular for $l$ being rectangular input patches and the $R$ resulting from an CNN with grid-aligned kernels.

For the case that the $R(l)$ are not all rectangular and $|\mathcal{L}|$ becomes large, checking Condition 3.2 can become prohibitively expensive. For this case, we derive a condition that corresponds to an upper bound on Condition 3.2 and can be evaluated in constant time with respect to $|\mathcal{L}|$:

**Condition 3.3.** $\min_{c \neq c_t} \sum_{i=1,j=1}^{w_{out},h_{out}} \Delta_{i,j,c} > 2R^{max}(\mathcal{L})$ with $R^{max}(\mathcal{L}) = \max_{l \in \mathcal{L}} |R(l)|$

*Proof.* $\Delta_{i,j,c} \leq 1$ implies $\sum_{i,j \in R(l)} \Delta_{i,j,c} \leq |R(l)| \leq R^{max}(\mathcal{L})$. For all $c \neq c_t$, using Condition 3.3:

$$\sum_{i,j \notin R(l)} \Delta_{i,j,c} = \sum_{i=1,j=1}^{w_{out},h_{out}} \Delta_{i,j,c} - \sum_{i,j \in R(l)} \Delta_{i,j,c_t} > 2R^{max}(\mathcal{L}) - R^{max}(\mathcal{L}) \geq |R(l)| \qquad \square$$

We note that Condition 3.3 corresponds to the condition proposed by Levine & Feizi (2020). It is, however, a strictly weaker condition than Condition 3.2. Thus, Condition 3.2 is preferable if all $R(l)$ are rectangular or $|\mathcal{L}|$ is of moderate size. We refer to Figure 5 in the supplementary material for an illustration of Condition 3.2 and 3.3.

## 3.2 MODEL

Crucially, the quality of the certification depends on $R^{max}(\mathcal{L}) = \max_{l \in \mathcal{L}} |R(l)|$: the larger this quantity becomes, the larger the left-hand side of Condition 3.2 or Condition 3.3 needs to be to fulfill the condition. We focus on the specific case where $f_\theta$ is realized by a convolutional neural network (CNN). In that case, $|R(l)|$ is determined fully by $l$ and the receptive field of the CNN. More specifically, we obtain $R(l) = \{(i,j) \mid \exists (\tilde{i}, \tilde{j}) \in l : |i - \tilde{i}| \leq \lfloor w_{rf}/2 \rfloor \wedge |j - \tilde{j}| \leq \lfloor h_{rf}/2 \rfloor \}$ for a receptive field size of $w_{rf} \times h_{rf}$ and ignoring operation strides.

Receptive field sizes of CNNs are determined by the shapes of the convolutional kernels as well as operation strides. We propose using standard CNN architectures such as ResNets but replacing most $3 \times 3$ convolutions by $1 \times 1$ convolutions, using stride 1 in (nearly) all operations, and removing all dense layers. This results in a network with very small receptive field sizes and thus small $R(l)$. We note that the proposed architecture is similar to BagNets (Brendel & Bethge, 2019) and using this type of model was concurrently proposed for certifying robustness against patch attacks by Zhang et al. (2020) and Xiang et al. (2020). BagNets obtain surprisingly high classification accuracy despite small receptive field sizes (Brendel & Bethge, 2019). Importantly, in contrast to BagNets, we do not apply a global average pooling on the final feature layer. This results in a dense output of shape $w_{out} \times h_{out} \times c_{out}$. The ratios $w_{in}/w_{out}$ and $h_{in}/h_{out}$ depend on the strides applied in the network and control mostly the computational overhead. We note that the cost for forward/backward passes in BagNets are in the same order of magnitude as those of a corresponding residual network. Because of the small receptive fields of BagNets, $|R(l)|$ is small if $l$ is a small contiguous region of the input, such as a rectangular patch.

We apply a Heaviside step function $H(x) = \begin{cases} 0, & \text{for } x < 0 \\ 1, & \text{for } x \geq 0 \end{cases}$ as final layer of $f_\theta$, which ensures $f_\theta(X) \in \{0, 1\}^{w_{out} \times h_{out} \times c_{out}}$. Similar to clipping Zhang et al. (2020) and masking (Xiang et al., 2020) this also ensures that a patch cannot flip the global classification by perturbing a local score so strongly that it dominates the globally aggregated score. However, since $H$ is constant nearly everywhere, it does not provide useful gradient information and thereby precludes end-to-end training. We address this by applying a "straight-through" type trick (Bengio et al., 2013) where we replace $H$ in the backward pass by its smooth approximation, the logistic sigmoid function $s(x) = \frac{1}{1+e^{-x}}$. That is, we use $H(x)$ in the forward pass but replace the true gradient of $H$ with $H'(x) := s'(x) = s(x)(1 - s(x))$. We explore alternatives to the Heaviside step function in Section A.3 in the appendix.

While the proposed model computes $f_\theta(X)$ in a single forward-pass and controls $|R(l)|$ indirectly via the architecture of $f$, we note that alternative models are compatible with BAGCERT. For instance, one could compute every element of the output $s_{i,j}$ via a separate forward pass of an arbitrary model on an ablated (Levine & Feizi, 2020) or cropped version of the input similar to Mask-DS-ResNet (Xiang et al., 2020). This also ensures that a specific element of the output depends only on the cropper/non-ablated part of the input. While these works are more flexible in terms of model architecture, they require a number forward passes proportional to the resolution of the output $s$, which would make inference (and end-to-end) training computationally much more expensive.

## 3.3 END-TO-END TRAINING

Having derived conditions that can be used for certifying robustness against patch attacks in Section 3.1 as well as differentiable model for the region scorer $f$ in Section 3.2, we now define a loss function for end-to-end training. We restrict ourselves to the case of a spatial sum aggregation $g_\sum$.

We recall Condition 3.3: $\min_{c \neq c_t} \sum_{i=1,j=1}^{w_{out}, h_{out}} \Delta_{i,j,c} > 2R^{max}(\mathcal{L})$. The corresponding loss for this can be defined as $L_H(\Delta, c_t, R^{max}) = H(\min_{c \neq c_t} \sum_{i=1,j=1}^{w_{out}, h_{out}} \Delta_{i,j,c} \leq 2R^{max})$, that is: the loss is 1 if there is a target class $c$ such that $\sum_{i=1,j=1}^{w_{out}, h_{out}} \Delta_{i,j,c}$ becomes smaller/equal two times the size of the maximum affected patch score region. However, this requires choosing $\mathcal{L}$ and the resulting $R^{max}(\mathcal{L})$ before training, which is undesirable. Instead, we stay agnostic with respect to the specific $\mathcal{L}$ and simply assume a uniform distribution[2] for $R^{max}(\mathcal{L})$, that is $R^{max}(\mathcal{L}) \sim \mathcal{U}(0, R)$. Here, $R$ corresponds to the maximum patch size (in region score space) we consider. This results in the loss

$$L_R(\Delta, c_t) = \int_0^R p(\tilde{R}) L_H(\Delta, c, \tilde{R}) d\tilde{R} = \int_0^R \frac{1}{R} H(\min_{c \neq c_t} \sum_{i=1,j=1}^{w_{out}, h_{out}} \Delta_{i,j,c} \leq 2\tilde{R}) d\tilde{R}$$

$$= 1 - \frac{1}{R} \int_0^R H(\min_{c \neq c_t} \sum_{i=1,j=1}^{w_{out}, h_{out}} \Delta_{i,j,c} > 2\tilde{R}) d\tilde{R}$$

$$= 1 - \frac{1}{R} \min(\frac{1}{2} \min_{c \neq c_t} \sum_{i=1,j=1}^{w_{out}, h_{out}} \Delta_{i,j,c}, R) = 1 - \frac{1}{2R} \min(\min_{c \neq c_t} \sum_{i=1,j=1}^{w_{out}, h_{out}} \Delta_{i,j,c}, 2R).$$

In practice, we minimize $\tilde{L}_R(\Delta, c_t) = - \min(\min_{c \neq c_t} \sum_{i=1,j=1}^{w_{out}, h_{out}} \frac{\Delta_{i,j,c}}{w_{out} \cdot h_{out}}, M)$ with $M = \frac{2R}{w_{out} \cdot h_{out}}$. This loss can be interpreted as a margin loss with margin $M$, where the margin corresponds to twice the maximum patch size in region score space against which we want to become certifiably robust.

**One-hot penalty** While we do not strictly enforce $\sum_c s_{i,j,c} = 1$, we sometimes found it beneficial to add a term in the loss that encourages $S = g_\sum(s)$ being approximately "one-hot", that is $L_{oh}(S) = \max_{c \neq c_{max}} S_c - S_{c_{max}}$ with $c_{max} = \arg\max_c S_c$. Since $S_c \in [0, 1]$, it holds that $L_{oh}(S) \in [-1, 0]$ and $L_{oh}(S) = -1$ iff $S_{c_{max}} = 1$ and $S_c = 0 \ \forall c \neq c_{max}$. The term $L_{oh}(S)$

---

[2]We note that other choices than the uniform distribution would be an interesting direction for future work, in particular if the defender has prior knowledge about more likely patch sizes and shapes.

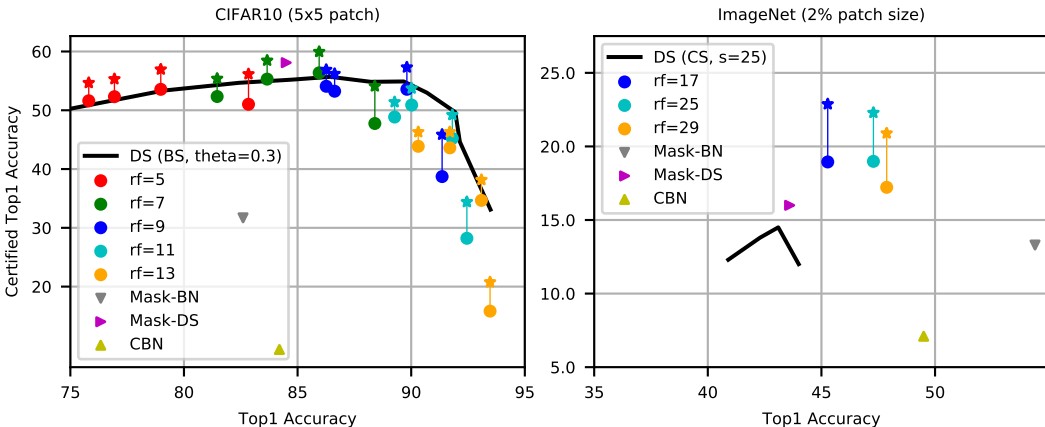

Figure 2: Clean versus certified accuracy on CIFAR10 and ImageNet for BAGCERT with different receptive fields and train margins ($M \in \{0.25, 0.5, 0.75, 1.0\}$ for CIFAR10, $M = 0.25$ for ImageNet) when certifying via Condition 3.3 (circles) and Condition 3.2 (stars), same setting connected by thin line. Smaller $M$ generally corresponds to larger clean accuracy for CIFAR10. Baselines are Derandomized Smoothing (DS) (Levine & Feizi, 2020), Masked BagNet (Mask-BN) and Masked DS-ResNet (Mask-DS) (Xiang et al., 2020), and Clipped BagNet (CBN) (Zhang et al., 2020). Results for these baselines are taken from the respective papers.

prevents training from prematurely converging to a solution where $s_{i,j,c}$ is approximately constant for all $i, j, c$, which we observed otherwise for tasks with many classes (e.g. ImageNet). The total loss becomes $L_{total} = \tilde{L}_R(\Delta, c_t) + \sigma L_{oh}(S)$, where $\sigma$ controls the strength of this one-hot penalty.

## 4 EXPERIMENTS

We perform an empirical evaluation of BAGCERT on CIFAR10 (Krizhevsky, 2009) and ImageNet (Russakovsky et al., 2015). We report clean and certified accuracy and compare to Interval Bound Propagation (IBP) (Chiang et al., 2020), Derandomized Smoothing (DS) (Levine & Feizi, 2020), Clipped BagNet (CBN) (Zhang et al., 2020), and PatchGuard (Xiang et al., 2020). For DS, we focus on block-smoothing and for PatchGuard, we focus on the masked BagNet (Mask-BN) because column smoothing for DS (and the derived Mask-DS for PatchGuard) perform poorly for non-square patches that are "short-but-wide" (see Figure 4). We notice that column smoothing and Mask-DS perform better than column smoothing and Mask-BN against square-patches; however, there is no reason an attacker should prefer square over non-square rectangular patches. Details on the BAGCERT model architecture and training can be found in Appendix A.1. Moreover, we focus on certified accuracy, a lower bound on the actual robustness of a model. Results for accuracy against a strong adversarial patch attack, corresponding to an upper bound on actual robustness, are discussed in Section A.2 in the appendix.

Figure 2 shows results for different methods against $5 \times 5$ patches for CIFAR10 corresponding to $2.4\%$ of the image size and patches of $2\%$ of the image size for ImageNet. For CIFAR10, when certifying accuracy via Condition 3.3, the Pareto frontier of BAGCERT follows closely the one reported for DS with block smoothing and $\theta = 0.3$. This is somewhat surprising given that both model and training procedure are very different and only the condition for certifying robustness is identical. We hypothesize that both approaches have reached close to optimal Pareto frontiers when certifying robustness via Condition 3.3. However, as Table 1 shows, BAGCERT requires (depending on its receptive field size) only between $39.0$ and $48.5$ seconds for certifying all $10.000$ test examples on a single Tesla V100 SXM2 GPU while DS with block smoothing requires 788 seconds. BAGCERT also clearly dominates Mask-BN and CBN, which utilize a similar model architecture, as well as IBP (not shown) which reaches $47.8\%$ clean and $30.3\%$ certified accuracy. Moreover, when applying Condition 3.2 for certification, certified accuracy is increased by approx. 3 percent points without changes in clean accuracy or any noticeable increase in certification time. In summary, the

| RF of BAGCERT | 5 | 7 | 9 | 11 | 13 | DS (BS) | DS (CS) |
|---|---|---|---|---|---|---|---|
| Certification time (seconds) | 39.0 | 40.6 | 43.2 | 45.9 | 48.5 | 788.0 | 28.0 |
| Number of parameters | 28M | 38M | 47M | 57M | 66M | 11M | 11M |

Table 1: Certification time for 10.000 CIFAR10 test examples and number of model parameters.

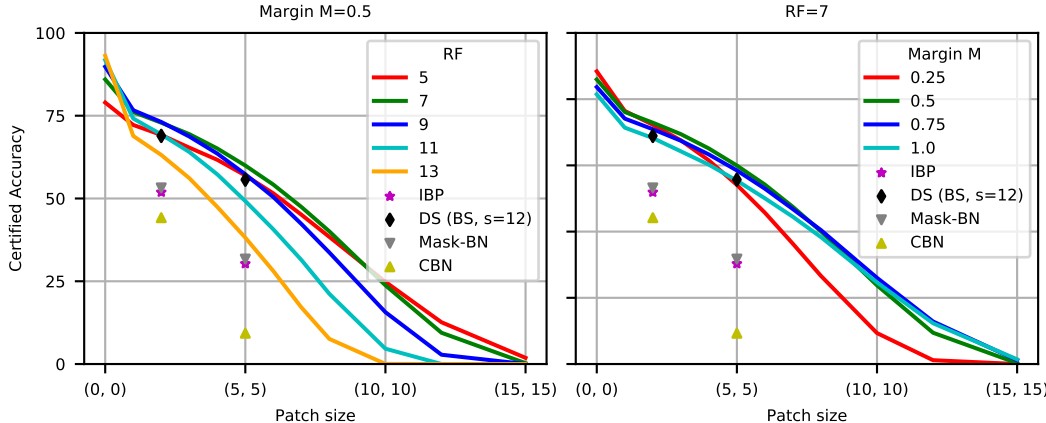

Figure 3: Certified accuracy against square patches of different sizes on CIFAR10. Shown is the performance for different receptive fields of BAGCERT (left) and train margins (right). Lines correspond to the same model (without retraining), evaluated against patches of different size.

strongest BAGCERT model with receptive field $7 \times 7$ and margin $M = 0.5$ can certify all 10.000 test examples in 43.2 seconds, reaching clean accuracy of $86\%$ and certified accuracy of $60\%$.

On ImageNet, BAGCERT also dominates all baselines in terms of certified accuracy, reaching $18.9\%$ via Condition 3.3 and $22.9\%$ via Condition 3.2 for receptive field size 17 and margin $M = 0.25$. Running certification for the entire validation set of 50.000 images takes roughly 7 minutes.

Figure 3 shows accuracy of BAGCERT certified via Condition 3.2 for square patches of different sizes on CIFAR10. Again, baselines are dominated for both $2 \times 2$ and $5 \times 5$ patches. Moreover, a single configuration of BAGCERT with receptive field size 7 and margin $M = 0.5$ performs close to optimal for all patch sizes and can certify non-trivial performance for up to $10 \times 10$ patch size. This implies that a single model can be used for a broad range of threat models. Figure 4 shows a similar analysis for non-square patches of a total size of 24 pixels. While BAGCERT with the same configuration as above achieves a certified accuracy of $40\%$ or more for any patch aspect ratio, performance of DS with column smoothing varies greatly with aspect ratio. In particular, "short-but-wide" patches of shape $24 \times 1$ or $12 \times 2$ reduce certified accuracy of column smoothing close to $0\%$. Since there is no reason to assume attackers will restrict themselves to square patches, we do not consider DS with column smoothing or Mask-DS (Xiang et al., 2020) general patch defenses, despite good performance for square patches and efficient certification according to Table 1.

## 5 CONCLUSION AND OUTLOOK

We have introduced a novel framework BAGCERT that combines efficient certification with end-to-end training for certified robustness. The main contributions are a model architecture based on a CNN with small receptive field, certification conditions that are applicable to a broad range of models, and a margin-loss based objective that is derived from the certification condition. The resulting model achieves high certified robustness against patches with a broad range of sizes, aspect ratios, and locations on CIFAR10 and ImageNet. Promising directions for future work are the exploration of other choices for the spatial aggregation function $g$ (such as ones using the "detect-and-mask" mechanism from PatchGuard (Xiang et al., 2020)) and corresponding certification conditions and losses that can be used for end-to-end training. Moreover, the development of alternative choices

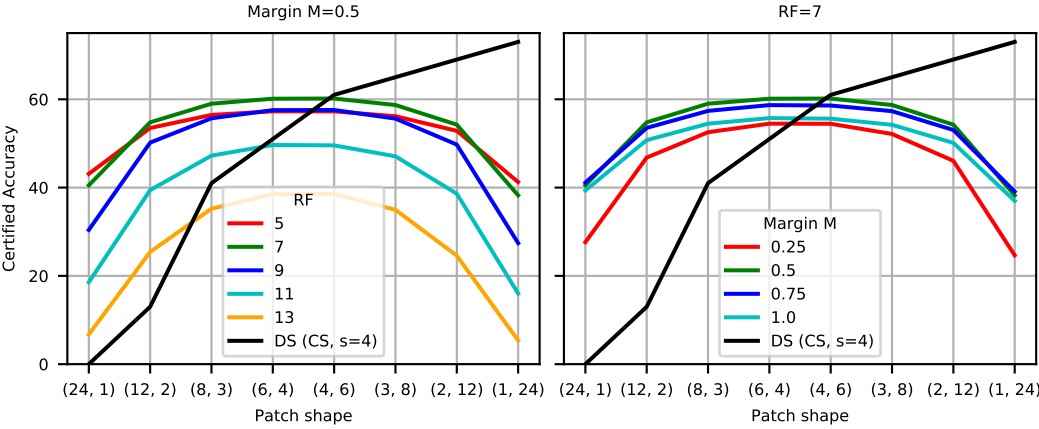

Figure 4: Certified accuracy against non-square patches of total size 24 pixels on CIFAR10. Shown is the performance for different receptive fields of BAGCERT (left) and train margins (right) compared to Derandomized Smoothing with Column-Smoothing (Levine & Feizi, 2020). Lines correspond to the same model (without retraining), evaluated against patches of different aspect rations.

for models with small receptive fields could be promising, such as ones based on learnable receptive fields or based on self-attention. Moreover, applying BAGCERT to other modalities than images would be an exciting avenue.

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

# A APPENDIX

## A.1 EXPERIMENTAL DETAILS

### A.1.1 CIFAR10

We use the following class of models for CIFAR10: we use a ResNet (He et al., 2016) base architecture, consisting of a single 3x3 convolution stem, followed by 8 residual blocks. We use stride one in all operations (that is: output resolution is $32 \times 32$) and use a constant width of 768 throughout the network. The last layer consists of a $1 \times 1$ convolution with 10 outputs. All layers use batch normalization (Ioffe & Szegedy, 2015) and ReLU. Blocks get assigned either a kernel size of 1 or 3, depending on the desired receptive field of the network. The following table summarizes the kernel-sizes of different blocks used in the experiments:

| RF of BAGCERT | stem | b1 | b2 | b3 | b4 | b5 | b6 | b7 | b8 |
|:---:|:---:|:---:|:---:|:---:|:---:|:---:|:---:|:---:|:---:|
| 5 | 3 | 3 | 1 | 1 | 1 | 1 | 1 | 1 | 1 |
| 7 | 3 | 3 | 1 | 3 | 1 | 1 | 1 | 1 | 1 |
| 9 | 3 | 3 | 1 | 3 | 1 | 3 | 1 | 1 | 1 |
| 11 | 3 | 3 | 1 | 3 | 1 | 3 | 1 | 3 | 1 |
| 13 | 3 | 3 | 3 | 3 | 1 | 3 | 1 | 3 | 1 |

Residual blocks use shake-shake regularization (Gastaldi, 2017) in the batch-wise mode. For residual blocks with kernel size 3, a special form of shake-shake regularization is used: the first residual path applies a $3 \times 3$ convolution followed by a $1 \times 1$ convolution, while the second residual path applies first a $1 \times 1$ convolution followed by a $3 \times 3$ convolution. This increases diversity of paths without changing the total receptive field of the network. Besides that, no additional regularization is applied, that is weight decay is $0.0$, and the one-hot penalty is set to $\sigma = 0.0$.

For training, we use the Adam optimizer with learning rate $0.001$, batch size 96, and train for 350 epochs. We apply a cosine decay learning rate schedule (Loshchilov & Hutter, 2017) with a warmup of 10 epochs. Moreover, we apply random horizontal flips and random crops with padding 4 for data augmentation.

### A.1.2 IMAGENET

We work on $224 \times 224$ inputs, which are extracted by rescaling the shorter side of the image to 256 pixels and extracting a random crop (training phase) or center crop (test phase) of size $224 \times 224$. Note that this input resolution differs from the $299 \times 299$ resolution used by Derandomized Smoothing Levine & Feizi (2020). In order to achieve comparable results, we evaluate against patches of size $32 \times 32$ ($32^2/224^2 \approx 2.04\%$) while DS test against patches of size $42 \times 42$ ($42^2/299^2 \approx 1.97\%$).

We use the following class of models for ImageNet: We use a ResNet base architecture, consisting of a single 3x3 convolution stem, followed by 8 residual blocks. We use stride 2 in blocks 1 and 3 and stride 1 otherwise (that is: output resolution is $56 \times 56$). We use width 64 in the stem and the first two blocks, width 128 in blocks 3 and 4, width 256 in blocks 5 and 6, and width 512 in blocks 7 and 8. The last layer consists of a $1 \times 1$ convolution with 1000 outputs. All layers use batch normalization and ReLU. Blocks get assigned either a kernel size of 1 or 3, depending on the desired receptive field of the network. The following table summarizes the kernel-sizes of different blocks used in the experiments:

| RF of BAGCERT | stem | b1 | b2 | b3 | b4 | b5 | b6 | b7 | b8 |
|:---:|:---:|:---:|:---:|:---:|:---:|:---:|:---:|:---:|:---:|
| 17 | 3 | 3 | 1 | 3 | 1 | 3 | 1 | 1 | 1 |
| 25 | 3 | 3 | 1 | 3 | 1 | 3 | 1 | 3 | 1 |
| 29 | 3 | 3 | 3 | 3 | 1 | 3 | 1 | 3 | 1 |

We apply neither shake-shake regularization nor weight decay. However, we set the one-hot penalty to $\sigma = 1.0$. For training, we use the Adam optimizer with learning rate $0.00033$, batch size 64, and train for 60 epochs. We apply a cosine decay learning rate schedule with a warmup of 10 epochs. Moreover, we apply random horizontal flips for data augmentation.

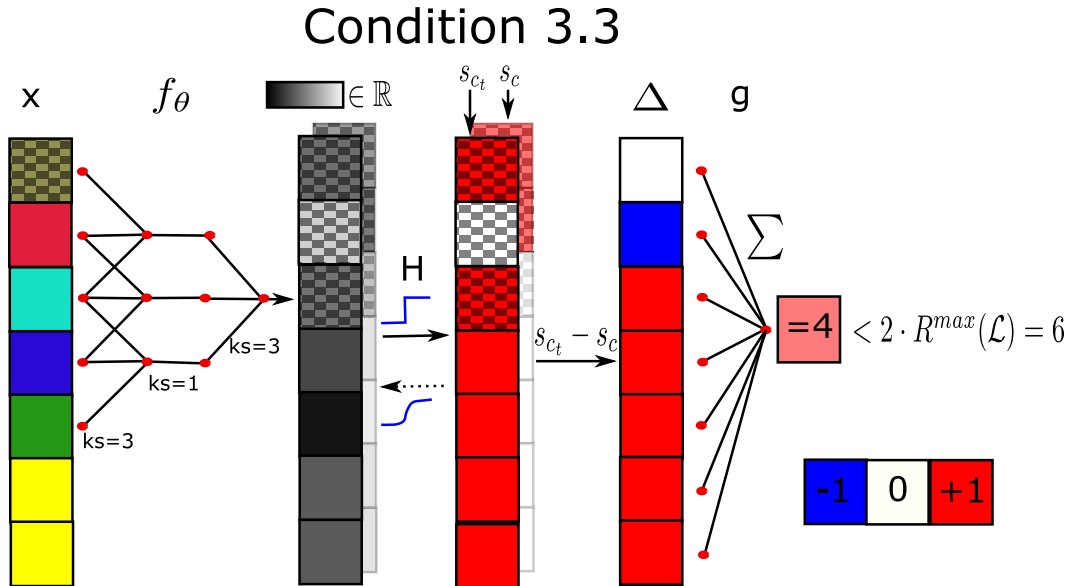

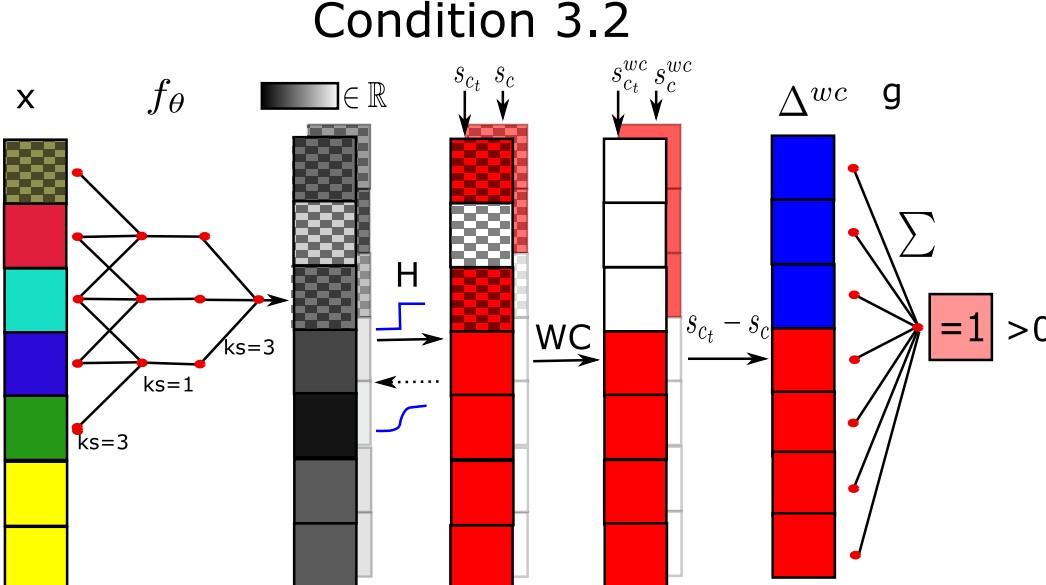

Figure 5: Illustration of BAGCERT certification for a 1D input and two classes. We assume for this example that $\mathcal{L}$ consists only of a single element; that is: the attacker can only place a patch at location $l = \{0\}$, shown by the checkerboard pattern in the input. The resulting $R(l)$ consists of the three top elements in region score space $s$ (shown again by a checkerboard pattern). Accordingly, $R^{max}(\mathcal{L}) = 3$. (Top) Certification via Condition 3.3: The regular network output $+4$ is compared to $2 \cdot R^{max}(\mathcal{L}) = +6$. Since $4 \leq 6$, the robustness of the prediction cannot be certified. (Bottom) Certification via Condition 3.2: region scores $s$ are replaced by $s^{wc}$ based on $R(l)$. The resulting network output is $+1$, which is greater than 0. Thus, robustness of prediction can be certified.

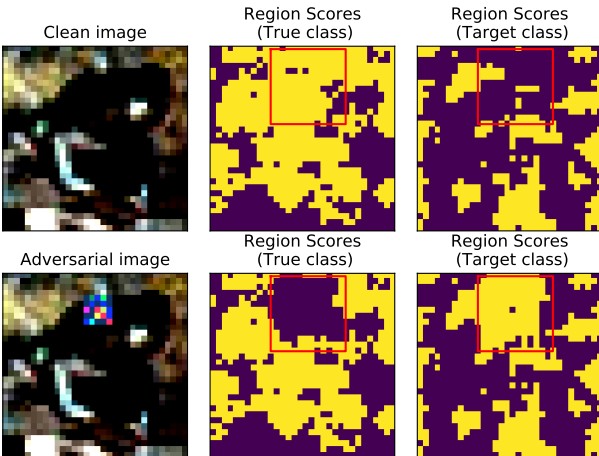

Figure 6: Illustration of an adversarial patch attack and its effect on region scores. Top row corresponds to the clean image (left), the resulting score maps for the true class (middle), and the score maps for the chosen target class (right). The bottom row shows the same for the image with an adversarial $5 \times 5$ patch inserted at the chosen region $l$. The red rectangle corresponds to $R(l)$.

### A.2 Robustness against Heuristic Patch Attack

While the certification conditions proposed in Section 3.1 allow computing a *lower bound* of a model's robustness against a specific type of patch attack, a model's true robustness against such attacks can be anywhere between this lower bound and the clean accuracy. In order to determine a tighter *upper bound* on robustness than clean accuracy, we perform a heuristic adversarial patch attack on the model and evaluate the model's accuracy on inputs that were modified by the attacker. Our threat model from Section 3 allows an attacker to place an arbitrary patch $p \in [0, 1]^{n \times c_{in}}$ at an arbitrary region $l \in \mathcal{L}$. We employ the following approach: we first select a region $l^* \in \mathcal{L}$ and target class $c^*$, and (once selected) keep this region and target class fixed and optimize the patch $p$ accordingly. Please note that no guarantee exists that actually the best region for an attack or the best patch are determined; thus, the resulting adversarial accuracy is only an upper bound.

Specifically, we focus in this evaluation on $5 \times 5$ square patches on CIFAR-10. Accordingly, $\mathcal{L}$ consists of all possible $5 \times 5$ subregions of a $32 \times 32$ input. Ideally, one would perform independent attacks at all possible regions $l \in \mathcal{L}$. However, this becomes quickly computationally intractable. We exploit specific design choices of BAGCERT to select one region and target class that may be particularly problematic for a model on an input assuming a spatial sum aggregation is applied. For this, we make directly use of Condition 3.2 and choose $l^*, c^* = \arg\min_{l,c} \sum_{i,j \notin R(l)} \Delta_{i,j,c}$. This choice corresponds to assuming a maximally effective patch attack that is able to achieve $\Delta_{i,j,c^*} = -1 \ \forall (i,j) \in R(l)$. A practical patch attack might not be able to achieve this ideal outcome (see also Figure 6) and thus $l^*, c^*$ are not necessarily optimal. However, they are reasonable choices that can be determined efficiently.

Once $l^*$ and $c^*$ are fixed, we perform a PGD attack (Madry et al., 2018) with 100 steps, a step size of 0.025, and the objective of maximizing the loss $\tilde{L}_R$ from Section 3.3 with margin $M = 1$. An illustration of such an attack is shown in Figure 6.

Figure 7 shows scatter plots of clean versus adversarial accuracy (left) and certified versus adversarial accuracy (right) for the BAGCERT models also shown in Figure 2. Interestingly, while clean and adversarial accuracy are highly correlated, the same does not hold true for certified and adversarial accuracy. In particular, adversarial accuracy seems to favor slightly larger receptive fields than certified accuracy. A potential reason for this can be seen in Figure 6: while a patch attack is typically effective for flipping the score of true and target class for the inner part of $R(l)$, it seems a lot harder to flip also scores close to the boundary of $R(l)$. For larger receptive fields of the model, this boundary effect seems to be amplified since the patch is smaller relative to the receptive field size.

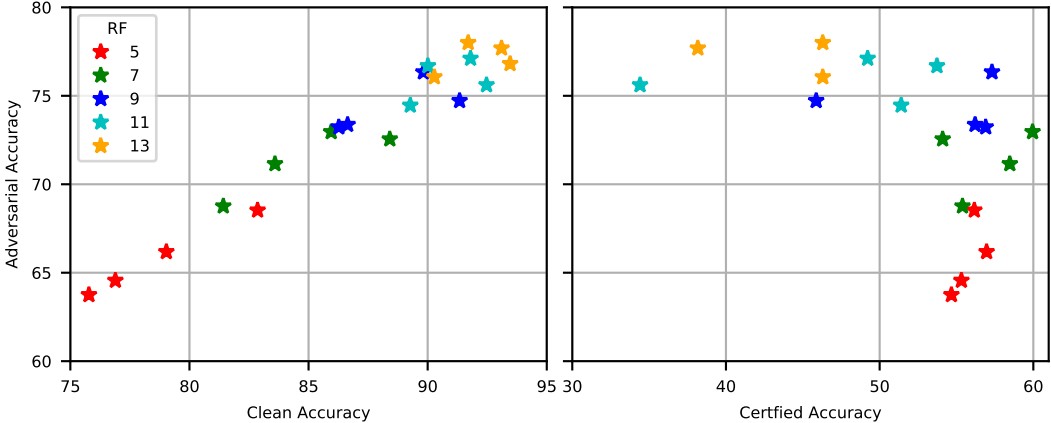

Figure 7: Scatter plots of clean versus adversarial accuracy (left) and certified versus adversarial accuracy (right). Color encodes different receptive filed sizes.

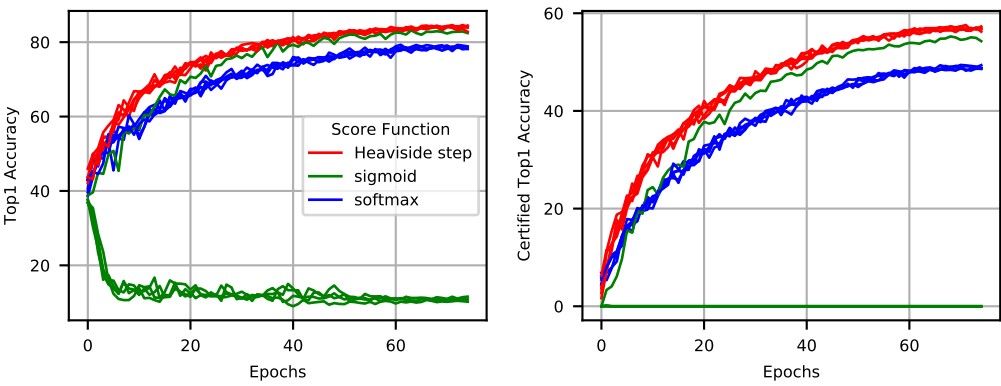

Figure 8: Comparison of Heaviside step function with alternative choices such as element-wise sigmoid or channel-wise softmax for five independent runs with different random seeds.

We consider reducing this gap between certified and adversarial accuracy further (that is: making bounds tighter) important future work. This will require both developing more effective attacks as well as improving certification procedures.

### A.3   EXPLORING ALTERNATIVES TO THE HEAVISIDE STEP FUNCTION

In this section, we explore alternative to the Heaviside step function as the last layer in the region scorer $f_\theta$ (see Section 3.2). For this, we relax region scores $s$ to be arbitrary values between $0$ and $1$, that is we have $s = f_\theta(x) \in [0,1]^{w_{out} \times h_{out} \times c_{out}}$. More specifically, we explore the element-wise sigmoid function $si(x_{i,j,c}) = \frac{1}{1+e^{-x_{i,j,c}}}$ and the channel-wise softmax function $sm(x_{i,j,c}) = \frac{e^{x_{i,j,c}}}{\sum_{c'} e^{x_{i,j,c'}}}$. Note that for the channel-wise softmax, $\sum_c s_{i,j,c} = 1$, while this is not enforced by element-wise sigmoid and Heaviside step function.

Figure 8 shows clean and certified accuracy (via Condition 3.2) on validation data during model training for five independent runs with the same configuration but different random seeds. The Heaviside step function ensure stable convergence to high clean and certified accuracy in all five runs (please note that performance is slightly lower than in Figure 2 because training was stopped after 75 epochs). The channel-wise softmax also shows consistent convergence, albeit to a lower level of performance. We attribute this to the hard constraint $\sum_c s_{i,j,c} = 1$ for region scores,

which is too restrictive for small and ambiguous regions. For the sigmoid function, one run reaches nearly the performance of the Heaviside step function while four other runs become unstable early during training and converge to chance level. Further hyperparameter tuning might be able to fix the convergence of models with the sigmoid function. However, we stick to the Heaviside step function since it provides stable performance without requiring further manual tuning.

