# OpenReview forum: "Efficient Certified Defenses Against Patch Attacks on Image Classifiers"
_ICLR.cc/2021/Conference — ICLR 2021 Poster_

### Official Review · AnonReviewer1 · 2020-10-23
**This paper presents a provable defense method called BAGCERT against patch attacks which uses an invariant of BagNet for certification.**

**Rating:** 6
**Confidence:** 4

**Review:**

This paper presents a provable defense method called BAGCERT against patch attacks which uses an invariant of BagNet for certification. By using the network with small receptive fields, this paper first analyzes the worst-case classification. The basic certification process is created by using a novel aggregation function. Finally, after using the same certification conditions as the Derandomized smoothing (Levine & Feizi, 2020), the certification could be evaluated within constant time. To further reduce the impact of the adversarial patch, the proposed method uses the certification condition as the objective loss to train the network. Empirical studies show the superiority of BAGCERT over other approaches.

Advantages:
1.The paper gives good formal descriptions and rigorous proofs.
2.The certification section is well structured. The narrative is logically layered and well-organized.
3. The paper gives a SOTA certified accuracy with high clean accuracy on ImageNet.

Concerns:
1. Overall, the paper uses essentially the same certification conditions as the Derandomized smoothing except that it replaces the network architecture with BagNet. In addition, the table that describes certification time shows that number of parameters of the proposed method is much larger than Derandomized smoothing. Is it fair to compare accuracy under networks with varying numbers of parameters that differ too much (38M:11M)?
2. Could you provide the training cost compared with Derandomized smoothing? In theory, the method is faster.
3. The description of R(l) in 3.1 is so unclear that it is difficult to understand the meaning of R(l).
4. The supplementary material should give some experimental results of the application of the proposed method to practical examples, for example, can you provide some results that uses some existing patch attack methods for evaluation?

---

> ### Author Response · Authors · 2020-11-13
> **Addressing the concerns**
>
> We are grateful for the helpful review and the constructive feedback. We will provide responses to the open points below:
>
> Concern 1a: "The paper uses essentially the same certification conditions as the Derandomized smoothing except that it replaces the network architecture with BagNet."
> We would like to clarify that only Condition 3.3 is equivalent to the condition in Derandomized smoothing. However, Condition 3.2 proposed by us is strictly stronger and applicable in typical cases (attacks on ConvNets with rectangular patches). Moreover, Condition 3.1 is a very general condition that can be used for settings not covered by Derandomized smoothing's condition. Moreover, our end-to-end training with a loss derived from the certification condition is an additional contribution over Derandomized smoothing.
>
> Concern 1b: "Is it fair to compare accuracy under networks with varying numbers of parameters that differ too much (38M:11M)?"
> We observe that for BagNets on CIFAR10 it is beneficial in general if they are "very wide" (compare Appendix A), more so than for standard ResNets used in Derandomized Smoothing, which might overfit severely if they would be as wide as our BagNets. So comparing different kinds of architectures is inherently difficult, because their "sweet spot" is typically at different number of parameters.
> We would like to note that on ImageNet, the best BagCert model has only 4.5M parameters, while the ResNet50 used in Derandomized smoothing has ~25M parameters. Figure 2 (right) shows that BagCert achieves superior performance on ImageNet  compared to Derandomized smoothing despite the model having ~6 times fewer parameters. Thus, competitive performance can also be obtained with smaller models in BagCert (probably larger BagNets would improve performance further on ImageNet).
>
> Concern 2: "Could you provide the training cost compared with Derandomized smoothing?"
> Thanks for raising this point. We would like to clarify that BagCert is more efficient than Derandomized block smoothing at inference time but not necessarily at training time. The reason is that BagCert performs end-to-end training with the certification process being part of training  while Derandomized smoothing only applies certification (the smoothing) post-hoc to a pretrained network. Thus, we do not claim that our method reduces training cost. But we also do not consider training time as relevant as inference time, as long as training remains practical and can scale to large problems such as ImageNet (which BagCert does). On ImageNet, training BagCert took between 4 and 5 days (depending on the receptive field size) on 4 Tesla V100 GPUs, which we consider practical. Levine and Feizi do not report training time for Derandomized smoothing, so it is hard to compare, but we expect a similar order of magnitude.
>
> Concern 3: "The description of R(l) in 3.1 is so unclear that it is difficult to understand the meaning of R(l)."
> We would add the additional sentence "Informally, $R(l)$ is the set of all indices of the score map that can be affected by a patch attack at region $l$, that is: the set of all outputs of $f_\theta$ whose receptive fields overlap with $l$.". We hope this sentence would clarify the meaning of $R(l)$?
>
> Concern 4: "Can you provide some results that uses some existing patch attack methods for evaluation?"
> We agree, heuristic attacks can be useful for providing a tighter upper bound on robust accuracy than clean accuracy. We are currently running an evaluation against patch attacks and will report results later.
>
> We will shortly upload a revised version of the paper. In case our response leaves questions open, we would be grateful for discussing these further.

---

> > ### Comment · AnonReviewer1 · 2020-11-24
> > **Response**
> >
> > I would like to thank the authors for providing additional explanations, and appreciate the authors' hard work.  After reading the other reviews, the authors response and the updated manuscript, I'm satisfied with the revised content.

---

### Official Review · AnonReviewer2 · 2020-10-27
**Convincing novel training and robustness certification approach for adversarial patches in image classifiers**

**Rating:** 7
**Confidence:** 3

**Review:**

This paper considers a problem of the defense against adversarial patch insertion attacks for image classification. Namely, it considers rectangular adversarial patches of fixed sizes and aspect ratios inserted in arbitrary locations of input images and requires from the desired model to obtain good classification performance on both clean and corrupted versions of datasets. Moreover, it is desired for results on the corrupted data to have certified robustness (theoretically guarantied classification performance given the model and the parameters of the attack).

To deal with this problem, this work uses a combination of ideas. Following prior work, it proposes to use CNN architectures with small receptive fields that compute class predictions for every spatial region in output feature maps, and to aggregate these predictions in a voting manner to obtain final class probabilities. The authors formally address the problem of certification and derive a novel stronger certification condition (compared to prior work), which allows to guarantee better performance for the proposed models. Moreover, the authors propose a novel training objective which is based on the certification condition, which allows for an end-to-end optimization of the model directly for the certified performance, in contrast to performing post-hoc adjustments to achieve certified robustness with pretrained models.

In practice, the proposed approach achieves superior certified performance on CIFAR10 and ImageNet datasets, while maintaining high clean accuracy. Additionally the certification process is reported to be about an order of magnitude faster compared to prior work. The approach is also shown to be more robust to rectangular adversarial patches.

Pros:
1) The paper is well-written and is pleasant to follow.
2) The proposed certification condition and training objective are novel and, as far as I can tell, are technically sound.
3) Formal analysis incorporates and properly relates the certification condition from prior work.
4) Experimental results are convincing and additionally include efficiency comparison.

Cons:
1) Might be not a con, but for me it is not clear which protocol is used for evaluations in experiments with varying patch sizes. Do I understand correctly that each point in figures 3 and 4 was obtained by retraining models from scratch with the chosen patch configuration? If that is the case, you can not imply that a single model is robust for different kinds of attacks, in that case it is the configuration of the model which is robust, but you will still need unique model parameters for any specific patch configuration.

Question:
1) Is it possible to apply the developed certification condition to prior work? It seems that prediction by region voting is the only requirement for applicability. If so, have you tried to use it on any prior works?
2) When you derive the objective from the certification condition 3.3 you assume that the size of the maximum affected patch score region is uniformly distributed. As far as I understand, this makes the objective robust to variations in patch sizes and geometry. In practice though, it is harder to certify robustness for larger stretched patches (Figures 3, 4). Does it make sense to consider distributions for $R^{max}(\mathcal{L})$ with densities shifted towards larger values?

Overall, I believe this is a strong paper, containing both theoretical and practical novel contributions and I think it should be accepted.

---

> ### Author Response · Authors · 2020-11-13
> **Clarification on Cons and Questions Raised**
>
> We would like to thank the reviewer for the thorough review and the constructive feedback. We will try to clarify the open points (and revise the paper accordingly):
>
> Con 1: "Do I understand correctly that each point in figures 3 and 4 was obtained by retraining models from scratch with the chosen patch configuration?"
> Lines in Figure 3 and 4 correspond to the same model, that is: there was _no_ retraining of models for different points of the same line. In particular, models were not trained for the specific patch size or aspect ratio. This implies that the same model parameters confer certified robustness to a wide variety of patch shapes.
> Thanks for raising this point; we will state this detail more clearly in the paper.
>
> Question 1: "Is it possible to apply the developed certification condition to prior work? It seems that prediction by region voting is the only requirement for applicability."
> The reviewer is correct, Condition 3.2 would be applicable to Derandomized Smoothing (Levine and Feizi) and we would expect similar benefits of using Condition 3.2 over Condition 3.3, that is an improvement of approx. 3 percent points of certified accuracy for Derandomized Smoothing. We see it actually as a strength of our work that individual contributions like Condition 3.2 are sufficiently general such that they are not restricted to BagCert but also applicable to other approaches.
> For PatchGuard (Xiang et al.), Condition 3.2 and 3.3 are not directly applicable because they assume a spatial-sum aggregation, while PatchGuard applies a detect-and-mask operation in the aggregation. Condition 3.1 only requires monotonicity; the detect-and-mask step in PatchGuard makes it (to our understanding) also non-monotonic. But one could potentially modify PatchGuard to using clipping instead of masking and by this make it satisfy the assumptions of Condition 3.1. This would be interesting future work; we will add this to the outlook of the paper.
>
> Question 2: "Does it make sense to consider distributions for $R^{max}(L)$ with densities shifted towards larger values?"
> Yes, it makes sense to consider different distributions for $R^{max}(L)$ if one has prior assumptions on typical patch sizes. In general, there is a trade-off between certified accuracy for small patches vs. certified accuracy for larger patches: this can be observed in Figure 3 (right) where the lines corresponding to different margins cross. So using "distributions for $R^{max}(L)$ with densities shifted towards larger values" will likely impair robustness against small patches (and clean accuracy). We use a uniform distribution because we want to stay agnostic with respect to the patch size (up to R) in this paper. But different choices for the distribution would be feasible, provided that they result in a closed form expression for the loss (or ones that can be efficiently approximated).
>
> We will shortly upload a revised version of the paper. In case our response leaves questions open, we would be grateful for discussing these further.

---

### Official Review · AnonReviewer5 · 2020-11-06
**Simple approach to verified robustness to patch attack producing good results.**

**Rating:** 7
**Confidence:** 3

**Review:**

Summary:
This paper deals with obtaining verified bounds on the accuracy of a model under attack restricted to patch modification (only a small, localized group of pixels can be modified). As opposed to previous methods (Chiang et al, 2020) which simply applied existing verification methods to the problem by enumerating possible patch locations, the proposed approach consist in modifying the structure of the network such that predictions are made densely and then aggregated. The dense prediction means that only some of the predictions can be affected by the adversarial patch, and so the certification process can use this to reason about the robustness at the aggregation level.
For network with small receptive fields, this will be particularly efficient (few predictions can be affected) so the authors propose to use CNN with small filter convolution in order to make the network more verifiable. Incorporating the verification procedure into the training objective is also described.

The proposed method achieves comparable or better results in terms of verified accuracy and nominal accuracy, while being significantly faster.

Comments:
- Figure 1 is quite helpful for understanding the principle of the proposed architecture.
- Why is it necessary to have the heaviside step function in the forward pass? Why wouldn't using the sigmoid function in the forward pass work? That way, the gradient used would actually corresponds to the objective.
- All the reasoning for robustness is here done at the aggregation phase, essentially enforcing that even flipping completely a prediction for a localized region does not change the global prediction. Could this be combined with the method of Chiang et al. to show that some regions can't be changed by more than a certain amount? This might allow the certification method to deal with networks with larger receptive fields?

Opinion:
The paper provides a simple, interesting approach, and describes it clearly. The empirical performance is also validated on both CIFAR and ImageNet.

---

> ### Author Response · Authors · 2020-11-13
> **Clarification on Heaviside Step Function and IBP**
>
> We would like to thank the reviewer for the time devoted to reviewing the paper and the constructive feedback. We will try to clarify the open points:
>
> 1) "Why is it necessary to have the heaviside step function in the forward pass? Why wouldn't using the sigmoid function in the forward pass work?"
> Thanks for bringing up this point. Actually, we conducted additional experiments (not yet contained in the paper) with alternative choices such as softmax and sigmoid. Softmax generally performed worse than the Heaviside step function. We attribute this to softmax enforcing  $\sum_{c} s_{i,j, c} = 1$, while the element-wise step function and sigmoid are more flexible. That seems to be helpful when dealing with small patches that can be intrinsically ambiguous.
> For the sigmoid function, we noticed that it can work just as well as the Heaviside step function in principle. However, we also noticed that in some runs it "stalled" on low accuracy levels on CIFAR10. We never observed this behaviour for the step function. It might be feasible to stabilize training with the sigmoid by careful hyperparameter tuning, but we sticked to the Heaviside step function which worked robustly for us. We will update the paper with according results.
>
> 2) "All the reasoning for robustness is here done at the aggregation phase, essentially enforcing that even flipping completely a prediction for a localized region does not change the global prediction. Could this be combined with the method of Chiang et al. to show that some regions can't be changed by more than a certain amount?"
> Thanks for this suggestion. From a quick glance, IBP would allow to have tighter bounds in $s^{wc}$ than the current trivial upper bound 1 and lower bound 0. These tighter bounds would then propagate to Conditions 3.1. and 3.2 and make these stronger. This sounds like a very promising follow-up for BagCert indeed. We will add this as a potential future direction to the paper, thanks again!
>
> We will shortly upload a revised version of the paper. In case our response leaves questions open, we would be grateful for discussing these further.

---

### Official Review · AnonReviewer4 · 2020-11-09
**Promising results on certifiable robustness to visible adversarial noise. Hard to decipher the level of technical contribution.**

**Rating:** 6
**Confidence:** 3

**Review:**

This paper investigates provable defenses to visible adversarial perturbations. Specifically the authors concentrate on defending against adversarial patches as introduced by Brown et al. [1] and not on other formulations of visible adversarial noise such as LaVAN (Karmon et al. [2]). The provable defense builds upon work by Xiang et al. [3], who utilise robust aggregation and BagNets to show the model can always recover correct predictions on certified images against any adversarial patch. The idea behind the defense is to constrain the receptive field size in convolutional layers; given a small adversarial patch and a large receptive field, the adversarial patch will be present in most extracted features, and so is more likely to change the model’s prediction. By limiting the size of the receptive field, the adversarial patch can only infect a small number of extracted features, after which the extracted features are binarized and robustly aggregated. The authors also introduce a new margin-based loss that encourages the model to be certifiable. Experiments with small adversarial patches on CIFAR-10 and ImageNet point to the efficacy of the approach. BagCert appears to outperform related certifiable approaches to patch attacks. The main boon of this approach lies in the fast certification time, since a constant number of forward-passes are required. The authors additionally have experiments showing the equivalence in robust accuracy against different shaped patches covering the same overall area size. Overall, I thought the paper was well-written, easy to follow and contains some interesting ideas. However, it is difficult to assess the level of technical contribution made here, since the core contributions in this work are also contained in Xiang et al [3].

As alluded to above, my main concern is the level of technical contribution made in this work. As far as I understand, this work is grounded in the ideas presented by Xiang et al [3] who use small receptive fields as a building block for robust classification. The main differences lie in the method of robust aggregation and that, here, the authors introduce a regularisation loss that encourages the model to be certifiable. As far as I could tell, a similar loss could have been introduced using the Xiang et al. approach and it is not specific to using the heaviside method of discretisation. It would be helpful if the authors please delineate the differences and contributions between this work and Xiang et al.? It is slightly disingenuous to say these works are concurrent contributions as stated in the contributions sections, since this work has been submitted ~4-5 months after Xiang et al. [3] was made publicly available.

Could this defense be applied to attacks that contain multiple, small localised patches (c.f. Karmon et al. [2])? I think the answer is probably yes, but it would be great to see an analysis in this direction.

The condition of a small receptive field implies an inherent trade-off between clean and robust accuracy. How should one tune the receptive field size to minimise this trade-off? This defense in its current formulation seems to be quite specific to image classification. Could this be extended to other modalities of data that use architectures that often extract global (instead of local) structure within data inputs (e.g. Transformers and NLP tasks)?

In the experiments, are the results compared against vanilla implementations of related approaches, or against approaches with relevant hyperparameters tuned? For example, when comparing with Xiang et al. [3], did you perform a sweep over receptive field sizes that minimise the clean-robust accuracy trade-off, or a sweep over the detection threshold, T?

As far as I understand, in Levine et al. [4], band smoothing can certify both column and row based patches (as opposed to square patches). In Figure 4, did the authors try to compare against row-based smoothing techniques (in addition to column smoothing)? I expect that this may solve the problem of zero certified accuracy on row patches. I encourage the authors to check this as it seems the Levine et al. [4] approach is outperforming BagCert for non-square adversarial patches.


[1] Brown, Tom B., et al. "Adversarial patch." arXiv preprint arXiv:1712.09665 (2017).

[2] Karmon, D., Zoran, D., and Goldberg, Y. Lavan: Localized and visible adversarial noise. arXiv preprint arXiv:1801.02608, 2018.

[3] Xiang, Chong, et al. "PatchGuard: Provable Defense against Adversarial Patches Using Masks on Small Receptive Fields." arXiv preprint arXiv:2005.10884 (2020).

[4] Levine, Alexander, and Soheil Feizi. "(De) Randomized Smoothing for Certifiable Defense against Patch Attacks." arXiv preprint arXiv:2002.10733 (2020).

---

> ### Author Response · Authors · 2020-11-13
> **Response to raised concerns and questions (Part 1/2)**
>
> We would like to thank the reviewer for the helpful review and the provided suggestions. We will try to clarify the open points:
>
> Concern 1: "Relation to Xiang et. al. (2020) and how fair is it to call our papers concurrent"
> We acknowledge that there is no consensus on what constitutes concurrent work. We started this project in March 2020 including the concept of using BagNets. The work of Xiang et al. appeared two months later (in May 2020) on a preprint server. Thus, we denote the work as concurrent, without intending to be "disingenuous" - after all, using BagNets (the main overlap between our work and  Xiang et al.) was something we settled on already before Xiang et al. got publicly available. We hope the reviewer agrees that this is a reasonable point of view - even in case the reviewer has a different opinion.
> In the case the reviewers and AC disagree and consider Xiang et. al. (2020) prior work, we still believe our work has sufficient novelty:
>  * We propose novel conditions for checking certifiability (Condition 3.1 and Condition 3.2) in Section 3.1
>  * We propose end-to-end training with a novel margin-based loss that trains the model to be certifiable (Section 3.3)
>  * Overlap with Xiang et. al. (2020)  is in the model (Section 3.2). However, even there we differ in terms of using a Heaviside step function for clipping (with a ”straight-through” type trick in the backward pass). Moreover,  we do not require upscaling CIFAR10 images to 192x192 as it is required in the PatchGuard, because we adapt BagNets to low resolution inputs.
>
> Concern 2: "Applying defense to multiple, small localized patches such as in LaVAN (Karmon et al.)"
> BagCert can also be applied to quantify robustness against multiple, small localized patches in principle. In this case, a feasible region $l$ would consist of several smaller local patches. One could define and compute $R(l)$ analogously to one connected patch. Condition 3.2 might become much more expensive computationally (because $R(l)$ would no longer be rectangular) such that one would have to resort to the weaker Condition 3.3. Moreover, we would expect that $R(l)$ would become larger for several local patches than for one connected patch of the same total size. Accordingly, we expect certified accuracy to decrease. An extreme case for this would be an L_0 threat model such as the one studied by Levine and Feizi in "Robustness Certificates for Sparse Adversarial Attacks by Randomized Ablation". In general, we see BagCert's main area of application on defending against adversarial patches as introduced by Brown et al., but it might be competitive in related but slightly different threat models as well.
>
> Concern 3: "How should one tune the receptive field size to minimize the trade-off between clean/certified trade-off accuracy?"
> This is a good remark. Generally, both the receptive field size and train margins M affect the trade-off and larger receptive fields and smaller margins increase clean accuracy at the expense of certified accuracy. Beyond that, one would have to specify what "minimise the trade-off" means since the problem is inherently multi-objective. Moreover, it is admittedly difficult to predict where on the Pareto frontier a network with a specific receptive field size and train margin would end up. This could be an interesting direction for future work.
>
> Concern 4: "Extension to other modalities? extract global (instead of local) structure within data inputs (e.g. Transformers and NLP tasks)"
> While we present BagCert for the case of 2d inputs and also perform our empirical evaluation on image data, the approach could easily be extended to other domains where ConvNets are applied such as 1d time series or 3d video data. For applying it to other kind of models such as e.g. Transformers, the main challenge would be to restrict all building blocks to act locally (e.g. only allow attention to nearby indices rather than global attention). We thank the reviewer for this remark; we think it can be a good direction for future work and it nicely fits the last sentence of our paper ("...the development of alternative choices for models with small receptive fields"). For problems which require inherently to extract global structure (or very long-range dependencies), our approach would not be an ideal choice in our opinion.

---

> ### Author Response · Authors · 2020-11-13
> **Response to raised concerns and questions (Part 2/2)**
>
> Concern 5: "Are the results compared against vanilla implementations of related approaches, or against approaches with relevant hyperparameters tuned? For example, when comparing with Xiang et al. [3], did you perform a sweep over receptive field sizes that minimize the clean-robust accuracy trade-off, or a sweep over the detection threshold, T? "
> For the models of Zhang et al., 2020 (Clipped BagNet)  and Xiang et al., 2020 (PatchGuard, Mask-BN and Mask-DS), we use performance numbers provided by the authors of the respective approach. We assume the authors have tuned the relevant hyperparameters appropriately.
>
> Concern 6: "Row/Column-based Derandomized Smoothing (Levine and Feizi) against non-square patches."
> We generally have that column-smoothing is strong against tall-but-narrow patches and weak against short-but-wide patches. For row-smoothing, it is exactly the opposite. Since the defender cannot know the aspect ratio chosen by the attacker in advance, both strategies are strictly weaker than a block-smoothing strategy that works well for all aspect ratios (BagCert's small and square receptive field is akin to block smoothing). An interesting question for future work would be if a kind of "ensembling" of different types of column/row/block-smoothing could outperform pure block smoothing for all aspect ratios.
> We would also like to note that also BagCert and other BagNet-based defenses can implement column- or row-smoothing by using mainly 1x3 or 3x1 kernels instead of 1x1 kernels. We did not explore this further but it could be an interesting direction for future work.
>
> We will shortly upload a revised version of the paper. In case our response leaves questions open, we would be grateful for discussing these further.

---

### Author Response · Authors · 2020-11-17
**Revised Version**

We have uploaded a revised version of our submission based on the comments by the reviewers. We would like to thank all reviewers again for their valuable input! Below we list the main changes compared to the initial submission:
 *  In Section A.2, we evaluate BagCert models against a practical patch attack to determine an upper bound on robustness as proposed by AnonReviewer1. The specific attack chooses the location of the patch based on certification condition 3.2 and performs a PGD attack to determine the respective patch.
 * We empirically compare and justify the choice of using a Heaviside step function in the forward pass versus using an element-wise sigmoid or channel-wise softmax, see Section A.3. We would like to thank AnonReviewer5 for this suggestion.
 * We simplify and clarify the definition of $R(l)$ as suggested by AnonReviewer1.
 * We mention that future work could investigate combining our method with the one of Chiang et al. as suggested by AnonReviewer5.
 * We state that results for baselines in Figure 2 are taken from the respective papers, clarifying a point by AnonReviewer4.
 * We state that lines in Figure 3 and Figure 4 correspond to the same model; clarifying a point made by AnonReviewer2.
 * We mention  in the outlook that applying/extending BagCert to other type of modalities or models would be a promising direction for future work, as proposed by AnonReviewer4.
 * Moreover, we mention in the outlook that covering other aggregation functions than spatial sum such as the ones proposed in prior work by Xiang et al. would be an exciting direction for future work, as discussed with AnonReviewer2.

We believe these changes improve the quality of the submission and would like to thank the reviewers again for their valuable suggestions and comments. If any points remain open, we would be grateful for discussing them further.

---

### Comment · ~Sravanti_Addepalli1 · 2021-05-11
**Request for Code**

Thank you for the interesting work! Could you please share your code for training and certification? This would be very helpful in being able to reproduce the results in the paper.

---

> ### Comment · ~Jan_Hendrik_Metzen1 · 2021-05-19
> **Your code request**
>
> Dear Sravanti, thanks for your interest! We haven't released code for this work and do not have plans for a code release in the near future, I am sorry. If you are facing challenges when reimplementing the method and reproducing its results, I am happy to offer support and provide feedback on open questions.
> Best,
> Jan

---

### Comment · ~Jan_Hendrik_Metzen1 · 2021-05-21
**Numbers for Figure 2**

Since there have been request regarding the exact numbers for the results shown in Figure 2 for BagCert, we are sharing them here:

 CIFAR10

|    |   Receptive Field |   Margin |   Clean acc. |   Cert. acc. (Cond. 3.3) |   Cert. acc. (Cond. 3.2) |
|---:|------------------:|---------:|-------------:|-------------------------:|-------------------------:|
|  0 |                 5 |     0.25 |        82.84 |                    51.02 |                    56.16 |
|  1 |                 5 |     0.5  |        78.98 |                    53.56 |                    56.96 |
|  2 |                 5 |     0.75 |        76.94 |                    52.31 |                    55.32 |
|  3 |                 5 |     1    |        75.81 |                    51.61 |                    54.67 |
|  4 |                 7 |     0.25 |        88.4  |                    47.75 |                    54.1  |
|  5 |                 7 |     0.5  |        85.95 |                    56.37 |                    59.95 |
|  6 |                 7 |     0.75 |        83.66 |                    55.3  |                    58.47 |
|  7 |                 7 |     1    |        81.46 |                    52.34 |                    55.39 |
|  8 |                 9 |     0.25 |        91.36 |                    38.72 |                    45.87 |
|  9 |                 9 |     0.5  |        89.81 |                    53.57 |                    57.31 |
| 10 |                 9 |     0.75 |        86.26 |                    54.08 |                    56.9  |
| 11 |                 9 |     1    |        86.63 |                    53.24 |                    56.21 |
| 12 |                11 |     0.25 |        92.45 |                    28.22 |                    34.41 |
| 13 |                11 |     0.5  |        91.81 |                    45.21 |                    49.22 |
| 14 |                11 |     0.75 |        90.02 |                    50.89 |                    53.73 |
| 15 |                11 |     1    |        89.27 |                    48.85 |                    51.4  |
| 16 |                13 |     0.25 |        93.47 |                    15.83 |                    20.76 |
| 17 |                13 |     0.5  |        93.09 |                    34.67 |                    38.17 |
| 18 |                13 |     0.75 |        91.7  |                    43.61 |                    46.3  |
| 19 |                13 |     1    |        90.31 |                    43.87 |                    46.3  |

 ImageNet

|    |   Receptive Field |   Margin |   Clean acc. |   Cert. acc. (Cond. 3.3) |   Cert. acc. (Cond. 3.2) |
|---:|------------------:|---------:|-------------:|-------------------------:|-------------------------:|
|  0 |                17 |     0.25 |       45.278 |                   18.946 |                   22.876 |
|  1 |                25 |     0.25 |       47.288 |                   18.986 |                   22.292 |
|  2 |                29 |     0.25 |       47.872 |                   17.222 |                   20.894 |

---

### Decision · Program_Chairs · 2021-01-07
**Final Decision**

**Decision:**

Accept (Poster)

**Comment:**

The paper develops a novel provable defense against patch-based adversasrial attacks on image classification system, by combining a novel architecture and certification procedure. The theoretical and experimental contributions are convincing and clearly advance the state of the art in provable defenses against adversarial perturbations.

The questions raised by the reviewers were addressed convincingly by the authors during the rebuttal phase, leading to unanimous consensus amongst reviewers towards acceptance. I recommend acceptance.